# Alterations in Proteostasis Mechanisms in Niemann–Pick Type C Disease

**DOI:** 10.3390/ijms25073806

**Published:** 2024-03-29

**Authors:** Iris Valeria Servín Muñoz, Daniel Ortuño-Sahagún, Christian Griñán-Ferré, Mercè Pallàs, Celia González-Castillo

**Affiliations:** 1Laboratorio de Neuroinmunobiología Molecular, Instituto de Investigación en Ciencias Biomédicas (IICB), Centro Universitario de Ciencias de la Salud (CUCS), Universidad de Guadalajara, Guadalajara 44340, Mexico; iris.servin@alumnos.udg.mx; 2Pharmacology Section, Department of Pharmacology, Toxicology and Therapeutic Chemistry, Faculty of Pharmacy and Food Sciences, Institute of Neuroscience, Universitat de Barcelona, 08028 Barcelona, Spain; christian.grinan@ub.edu (C.G.-F.); pallas@ub.edu (M.P.); 3Centro de Investigación Biomédica en Red (CiberNed), Network Center for Neurodegenerative Diseases, National Spanish Health Institute Carlos III, 28220 Madrid, Spain; 4Tecnologico de Monterrey, Escuela de Medicina y Ciencias de la Salud, Campus Guadalajara, Zapopan 45201, Mexico

**Keywords:** lysosome, lysosomal storage diseases (LSDs), cholesterol, protein degradation, protein folding

## Abstract

Niemann–Pick Type C (NPC) represents an autosomal recessive disorder with an incidence rate of 1 in 150,000 live births, classified within lysosomal storage diseases (LSDs). The abnormal accumulation of unesterified cholesterol characterizes the pathophysiology of NPC. This phenomenon is not unique to NPC, as analogous accumulations have also been observed in Alzheimer’s disease, Parkinson’s disease, and other neurodegenerative disorders. Interestingly, disturbances in the folding of the mutant protein NPC1 I1061T are accompanied by the aggregation of proteins such as hyperphosphorylated tau, α-synuclein, TDP-43, and β-amyloid peptide. These accumulations suggest potential disruptions in proteostasis, a regulatory process encompassing four principal mechanisms: synthesis, folding, maintenance of folding, and protein degradation. The dysregulation of these processes leads to excessive accumulation of abnormal proteins that impair cell function and trigger cytotoxicity. This comprehensive review delineates reported alterations across proteostasis mechanisms in NPC, encompassing changes in processes from synthesis to degradation. Additionally, it discusses therapeutic interventions targeting pharmacological facets of proteostasis in NPC. Noteworthy among these interventions is valproic acid, a histone deacetylase inhibitor (HDACi) that modulates acetylation during NPC1 synthesis. In addition, various therapeutic options addressing protein folding modulation, such as abiraterone acetate, DHBP, calnexin, and arimoclomol, are examined. Additionally, treatments impeding NPC1 degradation, exemplified by bortezomib and MG132, are explored as potential strategies. This review consolidates current knowledge on proteostasis dysregulation in NPC and underscores the therapeutic landscape targeting diverse facets of this intricate process.

## 1. Introduction

Proteostasis, also known as protein homeostasis, encompasses a series of essential steps throughout a protein’s functional lifespan. These steps encompass protein synthesis, post-translational modifications, proper folding, the preservation of correct folding, and eventual degradation [1]. The protein synthesis process involves the assembly of an 80S ribosome, a messenger RNA destined for translation into a protein, transfer RNA molecules, and, in certain instances, co-translational modifications like glycan incorporation, especially for specific proteins [2,3]. Folding depends on both protein-internal factors such as amino acid sequence and external effectors such as chaperones [4], and protein degradation can occur through various processes, mainly as the ubiquitin–proteosome system, autophagy, and lysosomal degradation [5,6]. Alterations in proteostasis have been noted in several diseases, including many neurodegenerative diseases such as Alzheimer’s, Parkinson’s, and Huntington’s diseases, as well as amyotrophic lateral sclerosis and even epilepsy, which progress as proteinopathies, i.e., there is an accumulation of abnormal proteins that evade the steps of proteostasis regulation and become toxic to the cells [7,8,9,10]. In Niemann–Pick type C, in addition to the dysregulation of proteins involved in NPC1 and NPC2 disease, the accumulation of some proteins upregulated in various neurodegenerative diseases, such as hyperphosphorylated tau protein, α-synuclein, and TDP-43, has also been noted, which is why it is called a proteinopathy [11,12,13,14].

NPC is a low-incidence disease estimated at 1:120,000 to 1:150,000 cases per live birth worldwide [15]. It is a disease belonging to the lysosomal storage diseases (LSD) group, with an accumulation of mainly unesterified cholesterol but also sphingomyelin, sphingosine, and gangliosides (GM2 and GM3). The accumulation of cholesterol in tissues becomes a systemic disease affecting various organs: the lung, spleen, pancreas, kidney, cardiovascular system, endocrine organs, gastrointestinal tract, peripheral nervous system (PNS), and central nervous system (CNS), with the liver and CNS being most affected [16]. In recent years, the evidence has increased with regard to the biosynthesis and transport of NPC1, which undergoes the same steps as other glycoproteins, including biosynthesis, co-translational glycosylation, proper folding in the ER, transport to the Golgi apparatus, processing into a mature complex glycosylated protein, and targeting to lysosomes for the dileucine motif in the C-terminus of the NPC1 protein [17].

All processes associated with the lifespan of NPC1 and NPC2 proteins hold significant importance for their functionality. Consequently, the deregulation of these processes carries implications for the pathophysiology of NPC disease. It has been determined that N-glycosylation of the NPC1 protein confers resistance to EndoH to avoid degradation. This glycosylation process facilitates the transport of the protein from the endoplasmic reticulum (ER) to the late endosomes/lysosomes (LE/Ly) [18]. Additionally, it has been observed that the loss of glycosylation in the N58 chain of the NPC2 protein hinders its transport into lysosomes [19]. The misfolding of NPC1 proteins, characteristic of various nonsense mutations, including the prevalent NPC1 I1061T mutation responsible for NPC, is recognized by the ERAD pathway as a misfolded protein, leading to degradation [20]. Consequently, the majority of therapies targeting NPC1 proteostasis primarily focus on improving protein folding. This includes the use of pharmacologic chaperones like abiraterone acetate, DHBP, and calnexin [21,22] or treatments aiming to modulate the expression of heat shock proteins, such as arimoclomol, JG98, AUY922, or the use of recombinant Hsp70 [23,24,25,26,27]. Changes in the mechanisms involved in the degradation pathways of proteins and other compounds due to the accumulation of cholesterol and other fatty acids in NPC have also been reported. For instance, the increased affinity of dynein for cholesterol-rich membranes keeps endolysosomes away from the periphery, preventing their recycling [28,29,30,31]. Moreover, sphingosine accumulation results in changes in the permeability of the lysosomal glycocalyx, leading to the release of proteases [32]. Additionally, the reduction in lysosomal calcium release due to sphingomyelin accumulation causes dysfunction in the fusion of lysosomes with early endosomes and autophagosomes, thereby impacting the degradation of components [33,34]. In this comprehensive review, we examine and deliberate upon the diverse mechanisms of proteostasis, where dysregulation has been documented, and the potential therapeutic avenues identified for addressing NPC disease. These mechanisms encompass N-glycosylation, the ERAD pathway, chaperone-assisted folding with heat shock proteins, autophagy, the lysosomal degradation pathway, and the ubiquitin–proteasome system.

### 1.1. N-Glycosylation, Implication in NPC1 Function

NPC1 is a glycosylated protein [3]. During its synthesis, immature glycans with high mannose content are co-translationally bound to the NH2 terminal [35]. NPC1 passes through the Golgi apparatus and the mature glycans through the action of glycan-modifying enzymes. It becomes resistant to the enzyme endoglycosidase (EndoH), which degrades glycans by cutting [18]. The mature WT NPC1 protein with such post-translational modifications has a half-life of 42 h. In contrast, the WT NPC1 protein without these post-translational modifications has a half-life of 9 h, and for the NPC1 protein with the I1061T mutation, this time is reduced to only 6.5 h [20].

This increased sensitivity to digestion with EndoH was confirmed in recent studies using cell lines with the human I1061T mutation in the NPC1 protein [36]. However, in mutant I1061T mouse models, the protein is highly resistant to EndoH, indicating species-specific differences. Resistance to EndoH was examined in both human NPC1 and mouse NPC1 with the R934L mutation, indicating that the same effect was observed. Mouse R934L-Npc1 showed more excellent resistance than human R934L-NPC1, suggesting that mouse Npc1 protein may be more efficiently transported to lysosomes than human protein [18].

This could be explained by the fact that adding glycans involves protein folding. In a 2022 analysis by Schultz et al., the probable glycan addition sites in a crystal structure of human NPC1 were assessed using NetNGlyc version 1.0 and GlycoEP https://bio.tools/glicoep (accessed on 25 February 2024) prediction software. It was found that three possible glycosylation sites are not present in mouse Npc1. The three glycosylation sites were inserted by a protein mutation that mimics the transport defects reported in human NPC1. This suggests two things: first, that this difference between species makes us question whether the mouse model is best suited to study this pathology, and second, and more importantly for this work, that these studies point to the importance of glycosylation sites in NPC1 for sensitivity to EndoH, leading to more significant degradation of the protein and thus impaired transport to lysosomes and a loss of function within these organelles [18].

The NPC2 protein is also a glycoprotein that is mainly localized in lysosomes, as it has a mannose-6-phosphate tag and is driven into lysosomes by mannose-6-phosphate receptors (MPRs) [37]. Regarding glycosylation, it has been shown that the Asn 58 residue is the most conserved glycosylated region in all NPC2 orthologs. If the protein lacks the N-link oligosaccharide at N58, then it cannot concretize its transport into lysosomes, and therefore, the NPC2 protein cannot fulfill its cholesterol-releasing function [19]. These studies show that the protein glycosylation involved in NPC, NPC1, and NPC2 plays a vital role in their co-localization in endolysosomes and, thus, in their function. In the case of NPC1, glycosylation protects against degradation during protein transport through the Golgi apparatus, and in the case of NPC2, the loss of the oligosaccharide chain of N58 prevents its transport to the lysosomes. From this, we can conclude that the post-translational processes of the NPC1 and NPC2 proteins are also of great importance for their functions and, therefore, also play a role in Niemann–Pick type C pathology.

### 1.2. ERAD Pathway and Chaperone Folding

The ER-associated degradation pathway (ERAD) may act as a quality-control step in selecting unfolded proteins for degradation. This process must include the steps of ejection of the misfolded protein into the cytoplasm, ubiquitin tagging, and degradation by the 26S proteasome [38]. Studies using NPC1 I1061T mutant human fibroblasts observed that this mutant protein was recognized by the ERAD system as a misfolded protein and subsequently degraded via the ubiquitin–proteasome pathway. However, this protein was not co-localized in lysosomes [20] Shultz et al. have found that the E3 ubiquitin ligase MARCH6 is responsible for regulating the labeling of NPC1 I1061T for degradation [36], and in this way, it controls the quality of protein folding (See Figure 1).

Some therapeutic alternatives to increasing NPC1 protein levels include improving protein folding through pharmacological chaperones or so-called pharmacological chaperone therapy, such as itraconazole. This azole derivative has been shown to bind to NPC1 in vitro and slightly improve cholesterol accumulation [39]. Also, the use of abiraterone acetate, which binds with high affinity to NPC1 in its N-terminal domain, resulted in an increase in NPC1 levels, improved co-localization in lysosomes, and a decrease in cholesterol accumulation [21]. Another therapeutic option that indirectly enhances the activity of ER-localized chaperones is the use of ryanodine receptor inhibitors, such as DHBP, which increase calcium stores in the ER by inhibiting these receptors that mediate calcium release from the ER. With this treatment, an increase in NPC1 I1061T protein was observed, its migration to lysosomes was promoted, and it regulated the accumulation of cholesterol and sphingolipids. These results are also consistent with the overexpression of calnexin, an important chaperone in protein folding dependent on calcium [22]. This shows how important the regulation of calcium concentration is for proper proteostasis, as some proteins responsible for folding are calcium-dependent. Moreover, this therapeutic approach shows promising results in regulating cholesterol accumulation, a key event in the pathophysiology of NPC. However, it is important to note that these types of therapies, as well as other pharmacologic therapies with chaperones, are only effective in nonsense mutations where protein folding is altered. However, these types of mutations account for the majority of cases of NPC [40,41].

### 1.3. Folding by Heat Shock Proteins

The activation of heat shock proteins (HSPs) or chaperones is mainly the response to misfolded proteins. These proteins help prevent neurodegenerative pathologies as well as some LSDs [42]. The main effectors of this process are the 70 kDA heat shock proteins (HSP70s), a group of proteins that respond to heat shock or cellular stress, which allows cells to gradually adapt to changes in the environment and survive under conditions of apoptosis and necrosis [43]. The HSP70s are ubiquitously expressed and are involved in all stages of protein synthesis, folding, maintenance, and degradation, thus playing a key role in proteostasis [44].

Studies on NPC have shown that the mutant NPC1 I1061T protein is functional in cholesterol trafficking but becomes unstable due to misfolding and is degraded via the ERAD pathway [20]. Based on this information, attempts have been made to find new potential therapeutic options, such as the use of arimoclomol. This small molecule induces HSP via the interaction between heat shock transcription factor 1 (HSF1) and heat shock elements (HSE) and has been identified as a potential treatment for several LSDs, such as Gaucher, Fabry, Sandhoff, and NPC diseases [24,25,42,45]. Specifically in NPC, treatment with arimoclomol has been shown to promote Hsf1 reactivation and normalize Hsp70 expression in Npc1−/− mice, leading to improved neurological phenotypes [24].

Another approach is the use of a recombinant HSP70 (rHSP70), with which a reduction in GLS species was observed in peripheral organs and the CNS of NPC mice, possibly due to the increase in sphingolipid-degrading enzymes by the action of rHSP70 [24]. The beneficial effects of using Hsp70 enhancement (rhHSP70) and bimoclomol, an analog of arimoclomol, were also observed at the stage of myelination in the cerebellum due to normalization of the mature oligodendrocyte population and rescue of atrophy in the cerebellum, resulting in improved behavioral outcomes [46]. The use of arimoclomol was evaluated in patients in a double-blind, randomized, placebo-controlled phase II/III clinical trial. A 65% annual reduction in the progression of NPC and a significant increase in HSP70 were observed with arimoclomol treatment [25].

In addition, the possibility that the allosteric inhibitor JG98 could promote the synthesis and folding of NPC1 by targeting the Hsp70 chaperone/cochaperone complex to disrupt the interaction between Hsp70 and BAG family co-chaperones was investigated and found to have an effect on correcting protein trafficking in 58 NPC1 variants [27]. The role of HSP90 was also evaluated through its inhibition in NPC1 I1061T mutant human fibroblasts. An elimination of cholesterol accumulation in endolysosomes and an increase in NPC1 protein expression were observed. In addition, it was observed that this inhibition increased the levels of HSP70 and HSP40. This overexpression also decreased cholesterol levels and increased NPC1 expression [26].

Heat shock protein family B member 1 (HSPB1) is another heat shock protein that is ubiquitously expressed, especially under stress conditions, and is involved in the refolding and degradation of unfolded polypeptides [47]. HSPB1 may be involved in delaying neurodegenerative diseases such as Alzheimer’s disease, as a transient interaction with early tau species has been observed, and this may delay the formation of neurofibrillary tangles [48]. Furthermore, in mouse models of NPC disease, the overexpression of Hspb1 was observed in delaying the progression of motor impairment and reducing the loss of Purkinje cells in the cerebellum, as well as in promoting neuronal survival in a cellular model [49].

## 2. Autophagy in NPC

Autophagy is a genetically regulated pathway for cellular components that mediate their degradation and thus participate in recycling, which is essential for proper cellular homeostasis. The autophagy process plays an important role in the constitutive degradation of long-lived proteins and some other macromolecules that cannot be degraded via the ubiquitin–proteasome pathway [50]. The steps of autophagy can be divided into induction, formation of the autophagosome, fusion with the lysosome, degradation of the cargo, and recycling of the autophagosome components [51]. Autophagy can be triggered by various stimuli, e.g., cellular stress, nutrient deprivation, hypoxia, protein aggregation, and others. The mammalian target of rapamycin complex 1 (mTORC1) mediates the initiation of autophagy through various pathways that dephosphorylate it and lead to the activation of molecular components involved in autophagy [52]. Neurons are particularly susceptible to accumulating unwanted cellular products, so proper regulation of the autophagy process is crucial. Indeed, it has been reported that alterations in autophagy may be associated with neurodegenerative diseases such as Alzheimer’s, Parkinson’s, and NPC [53]. As already known, the liver and CNS are among the organs most affected by NPC diseases. Regarding the role of autophagy in the liver, it has been described as being involved in recycling energy-rich substrates such as glycogen, proteins, lipids, and nucleic acids, which maintain reserves during starvation or cellular stress. Processes such as glycogen autophagy contribute to maintaining blood glucose homeostasis [54]. The first report of changes in the autophagy process in Npc1-deficient mice was described by Ko et al. They observed the presence of increased autophagic vacuoles in Purkinje cells and the accumulation of LC3 lipidation-associated proteins in the cerebellum [55].

However, increased autophagy has also been reported in NPC1-deficient human fibroblasts [50]. The upregulated autophagy is associated with Beclin-1, a protein involved in the autophagy process and lysosomal enzyme transport that interacts with the phosphatidylinositol 3-kinase (PI3K) complex for autophagosome formation. Indeed, it was observed that the expression of Beclin-1 was overexpressed in response to the induction of autophagy and that its inhibition causes a decrease in the degradation of long-lived proteins [50,56]. However, in addition to the reports of upregulation of autophagy, changes in the fusion process of late endosomes and autophagosomes have also been reported. For example, an inability to recruit components of the SNARE complex such as Vesicle Associated Membrane Protein 8 (VAMP8) and Vesicle Associated Membrane Protein 3 (VAMP3) has been observed in late endosomes with NPC1 protein deficiency, leading to an increase in steady-state autophagosomes that are unable to transport their cargo for degradation [57].

### Alterations in ER-Phagy

The ER has its own macroautophagy mechanism, ER-phagy [58]. In ER-phagy, areas of the ER are degraded by the formation of vacuoles that are degraded by lysosomes to regulate the size and activity of the ER in situations of cellular stress, unfolded protein response (UPR), and nutrient starvation, but it can also occur constitutively [58,59,60]. Constitutive and stress-induced ER-phagia share the same basic macro-autophagy machinery but differ in that constitutive ER-phagy does not require the starvation-induced receptors Atg39 and Atg40 [59]. The ER-phagy process requires the FAM134B proteins SEC62, RTN3, and CCPG1, which function as LC3 protein-binding receptors [61].

In cerebellar tissue lysates from NPC patients, the markers FAM134B, p62, and LC3- II were upregulated compared with controls, suggesting the dysregulation of this process in NPC. Overexpression of FAM134B has been linked to ER fragmentation [62], and mutations in this reticulophagy-regulating protein have been associated with the development of hereditary sensory and autonomic neuropathy type II (HSAN2), characterized by a reduction in sensitivity, particularly in the upper limbs, painless fractures, ulcer-mutilating complications, and osteomyelitis [63,64,65]. Alterations in the ER-phagy process have also been found in other neurodegenerative diseases such as Alzheimer’s and NPC, suggesting an important role at the neuronal level [36,63,64,66]. In NPC, it has been reported that one of the degradation pathways of NPC1 I1061T protein is ER-phagy, which is dependent on FAM134B. In the ER, the NPC1 I1061T protein is recognized as misfolded and begins to form autophagosomes by binding FAM134B to LC3 to fuse into lysosomes and be degraded subsequently [36,61]. These results suggest that ER-phagy is an important mechanism for regulating proteostasis in NPC, particularly the unfolded NPC1 protein caused by the most common mutation, NPC1 I1061T. In these cases, targeting one of the effectors of ER-phagy could be a therapeutic option.

## 3. Lysosomal Degradation Pathway

Lysosomes were discovered by Christian De Duve in 1950 and defined as organelles with a digestive function because they contain acid hydrolases inside them [67]. Lysosomes are involved in numerous functions, such as the digestion of endocytosed compounds, signal transduction, autophagy, and cellular homeostasis [68,69,70]. In particular, they play an important role in proteostasis, as they are responsible for the degradation of long-lived proteins, some misfolded proteins, and aggregated proteins [5]. Because lysosomes are involved in so many functions, alterations in these organelles are thought to have important implications for health, and pathologies associated with lysosomal dysfunction have been described, namely lysosomal storage diseases (LSDs), of which there are more than 70 [71].

In Niemann–Pick disease type C, which is the subject of this review, alterations were found at different levels of endocytosis and lysosomal degradation, which can be summarized as follows: alterations in endocytosis due to cholesterol accumulation, permeability of the lysosomal glycocalyx, and dysregulation of lysosomal calcium concentration (see Figure 2).

### 3.1. Alterations in Endocytosis Due to Cholesterol Accumulation in NPC

Endocytosis is the process by which internal membranes are formed from the plasma membrane that transport lipids and proteins from outside the cell or from the plasma membrane, which in turn are internalized into the cell [72]. The NPC1 protein is localized in late endosomes and lysosomes [73]. Late endosomes are transported through a network of microtubules with the help of the motor protein dynein, which moves the endosomes toward the negative end of the microtubules [74]. However, it has been observed that at high cholesterol concentrations, as in NPC disease, there are delays in the movement of endosomes [75,76]. For example, it has been observed that the pharmacological induction of cholesterol accumulation leads to changes in endosome distribution, possibly due to Rab protein dysfunction [77], particularly through the activity of the GTPase RAB7, whose involvement in the retrograde movement of late endosomes and lysosomes has been demonstrated, as it mediates the recruitment of motor proteins such as dynein by binding to the RAB7–RILP–ORP1L complex. ORP1L functions as a sensor of cholesterol levels, and RILP binds to the HOPS complex and to the p150 subunit of the dynein–dynactin complex [28,29,30,31]. In addition, defects in endosome transport have been observed following the overexpression of Rab7 GTPase, as endosomes accumulate in the perinuclear region in the same manner as cholesterol accumulation [28]. This is because Oxysterol-binding protein homologue (ORP1L) mediates the binding of the RAB7–RILP–ORPL1 complex to dynein at high cholesterol concentrations and does not allow the release of RAB7 binding mediated by the RAB7–RILP complex to the p150 subunit of the dynein–dynactin complex, since the process by which ER VAP (ER VAMP-associated protein) abolishes dynein binding occurs at low cholesterol concentrations, and late endosomes accumulate at the minus end of microtubules [29,31].

In addition, an accumulation of glycoproteins has been detected in NPC fibroblasts that occurs only in endosomes where cholesterol has accumulated and is not visible in endosomes with normal cholesterol concentration. Such accumulation could alter metabolic pathways in endosomes [78]. Another important component of the process of endocytosis is the SNARE proteins involved in the fusion of vesicle membranes with the membrane of the target organelle [79]. It has been shown that 11 of the SNAREs interact with cholesterol and are mainly found in late endosomes, recycling endosomes, and in the Golgi apparatus, so cholesterol may be important for SNARE localization [80]. One treatment that aims to regulate cholesterol levels by improving the solubility of hydrophobic compounds is 2-hydroxypropyl-β-cyclodextrin (HPβCD). However, this drug has been observed to lower cholesterol levels in a model of neuronal NPC cells but did not normalize the levels of NPC1 and [81]. In addition, adverse events such as hearing loss, fever, and meningitis have been reported in some patients [82]. HPβCD only crosses the blood–brain barrier in very small amounts, which is why high concentrations of this drug are required, or it is administered via a more invasive route for the patient, which increases the possibility of adverse events [83].

### 3.2. Alterations in Lysosomal Glycocalyx

The accumulation of sphingosine has been observed in mouse models as well as in NPC cell lines [32,84]. Sphingosine is a sphingolipid synthesized from ceramide and is considered an activator of apoptosis due to its properties of penetrating the lysosomal membrane and promoting the release of lysosomal hydrolases that damage the mitochondrial membrane. In contrast, phosphorylated sphingosine (S-1P) has been identified as a molecule that inhibits apoptosis [85,86]. In several cellular models of NPC, it has been observed that sphingosine accumulation causes changes in the plasma membrane and the fluidity of the lysosomal membranes [32]. Such accumulation of sphingosine in NPC may be due to the observed reduction in sphingosine kinase 1 (SphK1), which is one of the enzymes that phosphorylates sphingosine [87]. In addition, alterations in lysosome-associated membrane protein 1 (LAMP1), a protein with highly glycosylated luminal domains that contributes to the integrity of the lysosomal membrane and glycocalyx, have also been found in NPC [88,89]. In the cerebellum of Npc1 −/− mice, differences in LAMP1 glycosylation have been observed that are associated with disease progression and the death of Purkinje neurons [89]. Changes in LAMP1 glycosylation patterns have also been observed in studies with NPC1−/− cells, as they had high levels of mannose and sialylates, which can lead to changes in the lysosomal glycocalyx [35]. These changes in permeability could not only lead to dysfunction of lysosomal function and favor the aggregation of certain compounds but also cause the release of proteases that can damage other organelles, such as mitochondria, and trigger intrinsic apoptosis [15].

### 3.3. Dysregulation of Calcium Concentration

Lysosome represents an important Ca^2+^ reservoir that is involved in membrane fusion and fission events in this organelle associated with endocytic maturation, phagosome–lysosome fusion, lysosomal exocytosis, lysosomal reformation, vesicular transport, and signal transduction, so its proper homeostasis is crucial [90,91].

Alterations in lysosomal Ca^2+^ regulation have been found in neurodegenerative diseases such as Alzheimer’s disease, Gaucher disease, and NPC [92,93]. In NPC, some studies suggest a decrease in Ca^2+^ concentration in lysosomes [94]. However, it has been reported that there is not a decrease in lysosomal Ca^2+^ stores in NPC but rather a downward release of Ca^2+^ from lysosomes mediated by the receptor TRP mucolipin-1 (TRPML1). Insufficient Ca^2+^ release slows the fusion of lysosomes with endosomes and autophagosomes [95,96]. The TRPML1 receptor is a non-selective channel that allows Ca^2+^, Fe^2+^, and Zn^2+^ cations to exit and regulates several processes, such as lysosomal transport to the trans-Golgi network, fusion of phagosomes and lysosomes, reformation, and exocytosis of lysosomes [33,34]. It was observed that pharmacologically induced accumulation of sphingomyelin (SM) decreased TRPML1-mediated Ca^2+^ release, and SM degradation by the enzyme acid sphingo-myelinase (aSMase) restored TRPML1 activity [96]. SM is a sphingolipid that has been reported to be upregulated in Niemann–Pick type A, B, and C diseases [95,97,98]. TRPML1 may be an option as a therapeutic target in NPC because its overexpression reduces cholesterol accumulation, corrects lysosomal transport defects, and also has beneficial effects on other neurodegenerative diseases, such as Alzheimer’s disease, by enhancing dynein-dependent lysosomal transport by promoting the release of lysosomal Ca^2+^. This argues for the importance of this receptor and the release of lysosomal Ca^2+^ in lysosomal transport and neurodegenerative diseases [34,96,99].

## 4. Ubiquitin–Proteosome Pathway

The ubiquitin–proteosome system (UPS) mediates the degradation of most proteins, such as short-lived proteins and some misfolded proteins, and is involved in receptor regulation, assembly of multiprotein complexes, regulation of enzymatic activity, autophagy, DNA repair, and endocytic transport [6,23,100]. When the UPS is dysregulated, proteins can aggregate toxically in the cell [100]. This abnormal accumulation is associated with various neurological diseases, such as amyotrophic lateral sclerosis (ALS), Alzheimer’s disease, Parkinson’s disease, Huntington’s disease, and epilepsies [7,8,9,10]. Abnormal accumulation of proteins associated with some neurodegenerative diseases, such as hyperphosphorylated tau protein, α-synuclein, and TDP-43, has been reported in NPC [11,12,13,14]. These proteins are mainly degraded by the UPS [101,102,103]. The accumulation of such proteins in NPC may indicate an important dysregulation in UPS. Indeed, Cawley et al. recently found that ubiquitin C-terminal hydrolase-L1 (UCHL1) was overexpressed in the cerebrospinal fluid of patients with NPC, and that this was reduced in response to treatment with miglustat [88] UCHL1 is a protein that is mainly expressed in the brain and functions as a deubiquitinating enzyme, cleaving the binding of ubiquitin to ubiquitinated proteins [104,105]. In some diseases such as Alzheimer’s disease, the overexpression of UCHL1 appears to interact with the amyloid precursor protein (APP) and regulate the production of β-amyloid peptides. However, in NPC, its overexpression appears to be relevant to the pathogenesis of the disease [88,105].

As for the proteasome, various treatments that inhibit its degradation activity have been studied to increase the availability of NPC1 protein. For example, that treatment of fibroblasts with nonsense NPC1 mutations with MG132, a proteasome inhibitor, which was observed to increase the expression of NPC1 protein and favor its positioning within the endolysosomal compartment, improved the intracellular transport of cholesterol and M1(GM1) gangliosides [106]. Another treatment was the use of bortezomib, a proteasome inhibitor, which was used in fibroblasts from patients with NPC. After treatment, only a partial recovery of NPC1 protein levels was observed, but this recovery showed a significant reduction in cholesterol levels [107]. There are few studies showing alterations in the ubiquitin–proteasome system in NPC. However, the fact that this pathology shows an accumulation of several proteins, some of which are degraded via this pathway, raises the possibility that there are alterations in this labeling and degradation mechanism that have not yet been uncovered, and that their localization could be the result of an alteration in the ubiquitin–proteasome system.

## 5. Therapies Based on the Improvement of Proteostasis

As mentioned above, the deregulation of the mechanisms contributing to proper proteostasis is of great importance in Niemann–Pick disease type C. Therefore, the development of therapies based on the maintenance of this proteostasis has gained importance in recent years. Table 1 summarizes the different therapeutic options based on the regulation of proteostasis mentioned in this article.

## 6. Conclusions and Perspectives

The dysregulation of proteostasis constitutes a pivotal factor in numerous neurodegenerative disorders. It is a hallmark phenomenon of the aging process, which underlines its outstanding importance for the pathophysiology of these diseases. Within the context of Niemann–Pick disease type C (NPC), this review elucidates deviations in various proteostatic mechanisms, implicating NPC1 and NPC2 proteins. Notably, NPC manifests an accumulation of proteins associated with diverse neurodegenerative pathologies, including hyperphosphorylated tau protein, α-synuclein, and TDP-43 [11,12,13,14]. A plausible attribution to perturbations in the ubiquitin–proteasome system is suggested, given its paramount role in the degradation of these proteins. Despite limited research in this area, the lack of studies highlights the potential for increased investigation of possible alterations in the ubiquitin–proteasome system. This review also examines therapeutic interventions to modulate proteostasis. In particular, arimoclomol augments NPC1 protein folding through HSP transcription promotion. Although it is effective in reducing cholesterol accumulation and enhancing the transport of the mutant NPC1 protein, clinical trials reveal concerns regarding elevated creatinine levels, possibly due to the inhibition of organic cation transporter 2 [25]. Other FDA-approved treatments, such as valproic acid, chloroquine combination therapy, and abiraterone acetate, promise to mitigate cholesterol accumulation and enhance NPC1 transport to lysosomes. JG98, inhibiting the Hsp70–BAG family interaction, shows remarkable efficacy in correcting the transport of numerous NPC1 variants and reducing cholesterol accumulation.

Conversely, itraconazole shows lower efficacy and requires higher doses, probably due to its glycosylation inhibitory mechanism affecting the NPC1 protein, as demonstrated by Schultz et al. [18]. The proteasome inhibitor bortezomib inhibits the degradation of NPC1 but cannot correct the folding of NPC1. This underlines the urgent need to target therapeutic strategies’ synthesis and folding processes. In light of these findings, this review underscores the necessity to concentrate on therapeutic targets associated with protein lifespan events, particularly synthesis and folding, as a prospective avenue for significantly ameliorating the NPC pathology phenotype.

## Figures and Tables

**Figure 1 ijms-25-03806-f001:**
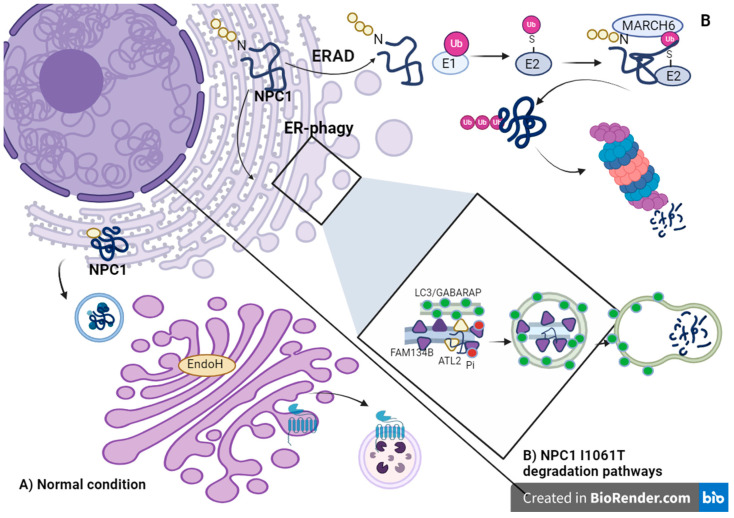
NPC1 protein transport and NPC1 I1061T degradation pathways. (**A**) Transport of NPC1 protein under normal conditions: After immature N-linked high mannose glycans are added to NPC1, NPC1 folds and is transported to the Golgi apparatus, where the glycans mature and provide protection that prevents it from being degraded by EndoH. The NPC1 protein is then labeled as a lysosomal protein in the trans-Golgi network with mannose-6-phosphate. NPC1 is localized in the lysosomal compartment by mannose-6-phosphate receptors. (**B**) I1061T NPC1 protein degradation pathways. The I1061T mutation leads to misfolding of the protein, and it has been represented as the protein that is degraded via two pathways: via the ERAD and pathway of ubiquitination, in which the protein is diverted to the cytosol, ubiquitinated, and transported to the proteasome for degradation, and via the FAM134B-dependent ER-phagy pathway, in which the protein is degraded after autophagocytosis in the lysosomes.

**Figure 2 ijms-25-03806-f002:**
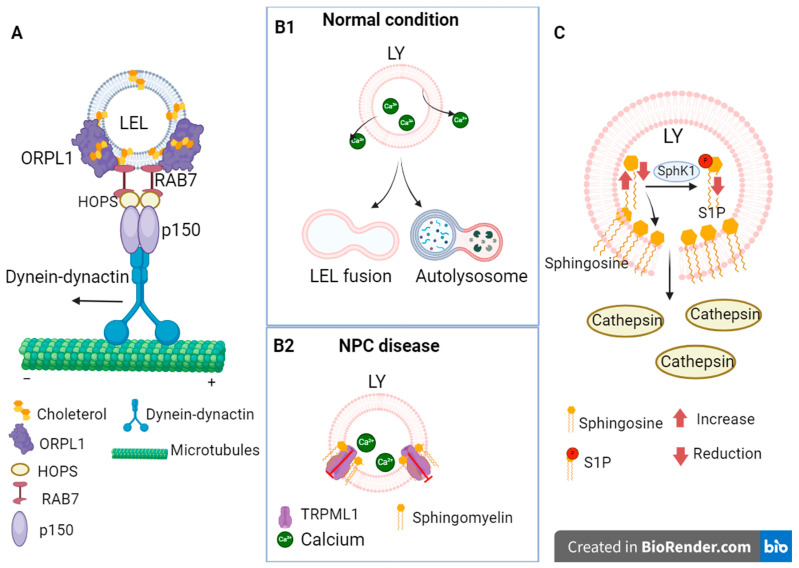
Changes in the process of endocytosis. (**A**) Retention of late endosomes at the minus end of microtubules due to the preference of the RAB7–RILP–ORPL1 and dynein–dynactin complexes for cholesterol-rich membranes. (**B1**) Under normal conditions, the TRPML1 channel facilitates the release of calcium from lysosomes, which favors the fusion of late endosomes with lysosomes and lysosomes with autophagosomes. (**B2**) In NPC, the increase in sphingomyelin blocks TRPML1 channels, impairing the release of lysosomal calcium. (**C**) Sphingosine accumulates in the lysosomes of NPC, leading to permeability of the lysosomal membrane and the release of enzymes such as cathepsins.

**Table 1 ijms-25-03806-t001:** Therapies regulating proteostasis in NPC.

Mechanism of Proteostasis in Which It Is Involved	Therapy or Therapeutic Target	Mechanism of Action	Effects	Considerations	Ref.
Synthesis	Valproic acid (VPA) and chloroquine	VPA reduces HDAC7 expression and increases NPC1 acetylation, and chloroquine neutralizes lysosomal hyperacidity.	Promotes folding and transport of NPC1-I1061T to LE/Ly, reduces cholesterol accumulation in NPC1-I1061T fibroblasts.	Valproic acid can cross the blood–brain barrier and can be administered orally. Treatment with VPA can alter the pH of the lysosome, so that combined treatment with chloroquine stabilizes the pH and has better effects.	[108]
Glycosylation	Itraconazole	Inhibits protein glycosylation; highly glycosylated proteins coat the lysosomal inner membrane, so the inhibition of glycosylation may cause increased permeability of the lysosomal glycocalyx, which increases the release of cholesterol into the lysosomes.	At high concentrations (10 μM), it significantly reduces cholesterol accumulation.	It has a slight effect on reducing cholesterol accumulation at low concentrations, possibly because it also inhibits the glycosylation of the NPC1 protein, which affects its function.	[39]
Folding	Abiraterone acetate	Interacts with NTD of NPC1 to mediate its folding.	Reduction in cholesterol concentration; upregulation and co-localization in lysosomes of NPC1 resistant to EndoH in three different mutations NPC1 E612D/P543Rfs*, I1061T, NPC1 Y394H/Y394H; these results show it as an effective treatment.	It is a drug approved by the FDA and the EMA for prostate cancer; its active metabolite is abiraterone; in animal models, it has been observed that it crosses the blood–brain barrier.	[21]
	DHBP	Inhibits ryanodine receptors and increases calcium levels in the ER; promotes the folding of NPC1 by calcium-dependent chaperones.	Increases levels of EndoH-resistant NPC1 I1061T, promotes the trafficking of NPC1 I1061T to lysosomes, and regulates the accumulation of cholesterol and sphingolipids.	With this treatment, it was observed that only the trafficking of a small fraction of NPC1 I1061T improved, but the reduction in cholesterol was significant.	[22]
	Calnexine	Mediates NPC1 I1061T folding.	Enhances trafficking of NPC1 I1061T to lysosomes and regulates the accumulation of cholesterol and sphingolipids.	Despite only increasing the transport of a low level of NPC1 I1061T NPC1, the reduction in cholesterol was significant.	[22]
	Arimoclomol	Co-induces heat shock proteins by stabilizing their interaction between heat shock factor (HSF-1) and HSES, which mediate the transcription of HSPs.	The treatment had significant effects on reducing the progression of NPC; increased HSP70 expression and reduced lipid accumulation.	Safety studies in patients with CPN show that the most common adverse event was vomiting (23.5%), and six of the patients presented increased serum creatinine.	[24,25]
	Recombinant human heat shock protein 70 (rhHSP70)	Amplification of HSP70 and therefore enhancement of sphingolipid-degrading enzymes.	Normalizes the oligodendrocyte population and rescues it from cerebellar atrophy; improved myelination in cerebellum; reduced GSL accumulation and improved NPC phenotype in a murine model.	In the study, treatment with rhHSP70 showed an improvement in myelination in the cerebellum; however, it is not known if the treatment is effective for increasing Npc1 levels and transport or if it is able to reduce cholesterol accumulation.	[46]
	Recombinant HSP70	Enhance NPC1 folding and enhancement of sphingolipid-degrading enzymes.	Reduced GM1 ganglioside accumulation; improved the motor phenotype in a murine model.	This treatment is not effective in knockout models, only in nonsense mutations.	[24]
	JG98 inhibitor	Disrupts the interaction between Hsp70 and BAG-1 and -3, which mediate the degradation of Hsp70 by the ubiquitin–proteasome system or the lysosomal autophagic pathway.	Correction in the trafficking of 58 NPC1 variants and reduction in the activation of SREBP-2, a transcription factor that regulates the expression of genes that participate in the synthesis of fatty acids.	This treatment showed significant responses even in variants with null or reduced activity of NPC1 transport and function; however, one of the disadvantages is that it is a fluorescent molecule, so that monitoring of cholesterol accumulation by filipin staining is not feasible.	[27,109]
	AUY922	Inhibition of HSP90, which when inactive mediates the release of HSF1 factor and promotes the expression of HSPs.	Overexpression of HSP70 and HSP40, increased transport of NPC1 from ER to LE/Ly and decreased cholesterol concentration.	The treatment with this drug for NPC has only been performed in vitro, but in the treatment of solid tumors it has been observed that the main adverse event is fatigue and decreased appetite.	[26,110]
	Overexpression of hspb1	Phosphorylated Hspb1 provides a neuroprotective effect by having anti-apoptotic functions.	In mice, there was delayed motor impairment and decreased Purkinje cell loss. In cells, promotes neuronal survival.	Hspb1 as a therapeutic target was only proven to be effective in reducing Purkinje cell loss in a murine model, but it was not evaluated whether there was a reduction in the accumulation of cholesterol or other fatty acids, so its effectiveness in this aspect cannot be determined.	[49]
Lysosomal pathway	Overexpression of TRPML1 receptor	Increasing its expression improves lysosomal calcium release, which is favorable for the fusion of LE and Ly and Ly with autophagosomes.	Enhances lysosomal trafficking and rescues it from cholesterol storage.	This treatment offers another approach that does not act directly on NPC1 or NPC2 proteins but improves lysosomal trafficking; also decreases cholesterol concentrations.	[96]
Degradation	Bortezomib treatment	Inhibits the 26s proteasome by inactivation of the chymotrypsin-like site in the proteolytic nucleus.	Partially increases NPC1 levels and reduces cholesterol content.	This treatment is not completely effective because it does not increase the co-localization of NPC1 to lysosomes despite showing a reduction in cholesterol accumulation.	[107,111]
	MG132	Inhibits the 26s proteasome	Increased NPC1 protein expression and co-localization within the endolysosomal compartment, reduction of GM1 accumulation, reduced cholesterol concentrations.	According to the study, it is not a safe treatment; at doses higher than 500 nM, there is reduction in cell viability; this treatment was shown to be effective in increasing NPC1 protein levels in fibroblast cell lines of six mutant variants.	[106]

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
