# Peer review of "Alterations in Proteostasis Mechanisms in Niemann–Pick Type C Disease"

_ijms, 2024, doi:10.3390/ijms25073806_

Round 1
Reviewer 1 Report
Comments and Suggestions for Authors
Dear the Editor
Munoz IVS et al comprehensively reviewed mechanisms involved in homeostasis of NPC proteins (ie NPC1 and NPC2). These authors discussed autophagy, lysosomal degradation pathway, and ubiquitin-associated pathways play major roles, while N-glycosylation, ER-associated degradation, heat shock protein-mediated folding might further modulate NPC protein homeostasis. Finally, several therapeutic strategy was discussed based on these molecular basis. Overall, this review was well-organized. Readability of this narrative review seemed fine.
This study provided a possible mechanism of NPC protein homeostasis under pathophysiological conditions. Classically, the protein level of NPC1 and NPC2 have been considered by either lysosomal degradation or ubiquitin-dependent mechanism. These authors further provided a possibility of autophagy-dependent mechanism. The latter is a novel hypothesis that fills the scientific gap of previously unappreciated mechanism of NPC.
Comprehensive discussion for NPC protein homeostasis is rarely seen in other publication.
Conclusions were consistent with these author's discussion made in this manuscript.
References are appropriate.
Figures were well-prepared.
Major concerns:
None.
Minor concerns:
1) In L40, 80s should be 80S.
2) In 262, please explain HSAN II.
Author Response
Reviewer 1
We thank Reviewer #1 for his/her most favorable comments on our work.
Minor concerns:
1) In L40, 80s should be 80S.
2) In 262, please explain HSAN II.
Response: Both corrections were done, and one reference was added (between references 37 and 38).
Reviewer 2 Report
Comments and Suggestions for Authors
- The paper provides a comprehensive review of the alterations in proteostasis mechanisms in Niemann-Pick type C disease and the potential therapeutic strategies to address them. However, the paper would benefit from some revisions and clarifications to improve its quality and readability.
- The authors should discuss the therapies listed in Table 1 in relation to the proteostasis pathways they target. For example, how do arimoclomol, abiraterone acetate, and JG98 affect the ERAD pathway, the chaperone folding, and the Hsp70 complex, respectively? How do these therapies compare with each other in terms of efficacy, safety, and specificity? This would help the readers to better understand the rationale and the implications of these therapeutic approaches.
- Figure 2 should be revised to correct the labeling of the panels. There are two panels labeled as "A", which is confusing. The authors should also explain the meaning of the colors and symbols used in the figure legend.
- The authors should add a conclusion section that summarizes the main findings and contributions of the paper, as well as the limitations and future directions for research in this field. This would help to highlight the significance and the impact of the paper and to provide some guidance for further studies.
Author Response
Reviewer 2
We thank Reviewer #2 for his/her favorable comments on our work.
Comment 1. The authors should discuss the therapies listed in Table 1 in relation to the proteostasis pathways they target. For example, how do arimoclomol, abiraterone acetate, and JG98 affect the ERAD pathway, the chaperone folding, and the Hsp70 complex, respectively? How do these therapies compare with each other in terms of efficacy, safety, and specificity? This would help the readers to better understand the rationale and the implications of these therapeutic approaches.
Response: Thank you for the accurate and relevant comment. To address the reviewer's request, we have modified our original Table 1. We now start with the classification according to the mechanism of proteostasis in which the treatment is involved, followed by the therapeutic target, mechanism of action, effects and considerations (including the aspects mentioned by the reviewer where possible). We hope that this new table will be more informative for readers to better understand the rationale and implications of these therapeutic approaches, as the reviewer very correctly noted. (See new Table 1 in the manuscript).
Comment 2. Figure 2 should be revised to correct the labeling of the panels. There are two panels labeled as "A", which is confusing. The authors should also explain the meaning of the colors and symbols used in the figure legend.
Response: The reviewer's comment is correct, and we are grateful that this error has been detected. The correction has been made in the new Figure 1 (see new figure 1 in the R2 version of the manuscript).
Comment 3. The authors should add a conclusion section that summarizes the main findings and contributions of the paper, as well as the limitations and future directions for research in this field. This would help to highlight the significance and the impact of the paper and to provide some guidance for further studies.
Response: We thank the reviewer for the opportunity to address this aspect. In accordance with the reviewer´s request, we have added a whole new section 6 with conclusions and perspectives. (See new section 6, for conclusions and perspectives).
Reviewer 3 Report
Comments and Suggestions for Authors
The review by Servin Munoz et al. presents an interesting position about the managment of the Niemann-Pick disease by intervening on proteostasis alterations. The concept is not new but it is sound and worth exploring in our opinion. The paper should therefore be useful to researchers in the field.
However, in its present form, the review still lacks some formal and conceptual elements.
The abstract lacks a brief summary of the proposed therapeutic interventions, conclusions are not anticipated and the interest of the reader could therefore be soon lost.
The introduction could be re-organised by expanding all the concepts anticipated in the abstract, such as a general framing of the N-P disease (cholesterol accumulation..), of proteostasis disruptions, of current interventions and the proposed ones.
Also in the introduction, a general framing encompassing all the points later explained must be provided.
In the section dedicated to therapies, Table 1 is interesting but it is unclear which one of the types of treatment they cited is applicable and advised by the authors in the specific condition object of the review.
Seminal papers such as Subramaniam et al. JBC 2020, 295, 23 have not been cited and could possibly help in the conceptualisation of an eventual therapy.
Comments on the Quality of English Language
English in understandable and generally fine but a professional mother language help could improve the prose which is evidently not very fluent
Author Response
Reviewer 3
We thank Reviewer #3 for his/her favorable opinion about our work.
Comment 1. The abstract lacks a brief summary of the proposed therapeutic interventions, conclusions are not anticipated and the interest of the reader could therefore be soon lost.
Response: We thank the reviewer for pointing this out and appreciate the opportunity to correct this aspect. In accordance with the reviewer's comment, we have rewritten the abstract and added a brief summary of the proposed therapeutic interventions and emphasized the purpose of the review.
Comment 2. The introduction could be re-organised by expanding all the concepts anticipated in the abstract, such as a general framing of the N-P disease (cholesterol accumulation), of proteostasis disruptions, of current interventions and the proposed ones.
Response: At the reviewer´s suggestion, we have expanded the information in the introduction to establish a link between disturbances in proteostasis and the pathophysiology of NPC disease.
Comment 3. Also in the introduction, a general framing encompassing all the points later explained must be provided.
Response: Included in the changes to the introduction
Comment 4. In the section dedicated to therapies, Table 1 is interesting but it is unclear which one of the types of treatment they cited is applicable and advised by the authors in the specific condition object of the review.
Response: This comment is related to comments 1 and 3 of reviewer #1. To amend this, we have modified Table 1 and added a new final section with conclusions and perspectives, in which we further elaborate on the types of treatment, their target, mechanisms of action, effects, and specificities. In the final section, we also highlight the main therapeutic interventions to modulate proteostasis and the necessity for further research.
Comment 5. Seminal papers such as Subramaniam et al. JBC 2020, 295, 23 have not been cited and could possibly help in the conceptualisation of an eventual therapy.
Response: We thank the reviewer for bringing this important reference to our attention. We have, therefore, added the information from this reference in Table 1. Finally, we would like to thank the three reviewers for their favorable opinions and for the time that they generously devoted to reviewing our manuscript, as well as for their accurate review and comments that undoubtedly improved the precision of our manuscript. We do hope that we have clarified the points indicated.
Round 2
Reviewer 2 Report
Comments and Suggestions for Authors
The paper provides a thorough review of proteostasis mechanisms in Niemann-Pick Type C (NPC) disease, covering alterations from synthesis to degradation. The detailed discussion on therapeutic interventions is particularly commendable.
The exploration of pharmacological aspects of proteostasis in NPC, including the use of Valproic acid and protein folding modulation treatments, offers valuable insights for future research and clinical applications.
The structure of the paper, with clear delineation of the proteostasis process and its dysregulation in NPC, aids in the understanding of this complex topic.
Consider expanding on the implications of species-specific differences in NPC1 protein trafficking for therapeutic response. This could provide a more nuanced view of the challenges in translating research from model organisms to humans.
Overall, the paper is well-written and informative, providing a significant contribution to the field of lysosomal storage diseases. The manuscript is suitable for this journal.